# Towards Feasible Solutions for Load Monitoring in Quebec Residences [note 1]

**DOI:** 10.3390/s23167288

**Published:** 2023-08-21

**Authors:** Sayed Saeed Hosseini, Benoit Delcroix, Nilson Henao, Kodjo Agbossou, Sousso Kelouwani

**Affiliations:** 1Department of Electrical and Computer Engineering, Laboratoire d’innovation et de Recherche en Énergie Intelligent, Hydrogen Research Institute, Université du Québec à Trois-Rivières, Trois-Rivières, QC G8Z 4M3, Canada; nilson.henao@uqtr.ca (N.H.); kodjo.agbossou@uqtr.ca (K.A.); 2Hydro-Quebec Research Institute, Laboratoire des Technologies de l’énergie d’Hydro-Québec, Shawinigan, QC G9N 0C5, Canada; delcroix.benoit@hydroquebec.com; 3Department of Mechanical Engineering, Laboratoire d’innovation et de Recherche en Énergie Intelligent, Hydrogen Research Institute, Université du Québec à Trois-Rivières, Trois-Rivières, QC G8Z 4M3, Canada; sousso.kelouwani@uqtr.ca

**Keywords:** Non-Intrusive Load Monitoring (NILM), low-sampling load disaggregation, statistical analysis, machine learning algorithms, electric baseboards, electric water heaters

## Abstract

For many years, energy monitoring at the most disaggregate level has been mainly sought through the idea of Non-Intrusive Load Monitoring (NILM). Developing a practical application of this concept in the residential sector can be impeded by the technical characteristics of case studies. Accordingly, several databases, mainly from Europe and the US, have been publicly released to enable basic research to address NILM issues raised by their challenging features. Nevertheless, the resultant enhancements are limited to the properties of these datasets. Such a restriction has caused NILM studies to overlook residential scenarios related to geographically-specific regions and existent practices to face unexplored situations. This paper presents applied research on NILM in Quebec residences to reveal its barriers to feasible implementations. It commences with a concise discussion about a successful NILM idea to highlight its essential requirements. Afterward, it provides a comparative statistical analysis to represent the specificity of the case study by exploiting real data. Subsequently, this study proposes a combinatory approach to load identification that utilizes the promise of sub-meter smart technologies and integrates the intrusive aspect of load monitoring with the non-intrusive one to alleviate NILM difficulties in Quebec residences. A load disaggregation technique is suggested to manifest these complications based on supervised and unsupervised machine learning designs. The former is aimed at extracting overall heating demand from the aggregate one while the latter is designed for disaggregating the residual load. The results demonstrate that geographically-dependent cases create electricity consumption scenarios that can deteriorate the performance of existing NILM methods. From a realistic standpoint, this research elaborates on critical remarks to realize viable NILM systems, particularly in Quebec houses.

## 1. Introduction

Home Energy Management Systems (HEMSs) are the key enabler of the residential demand-side role in power system flexibility towards the energy transition [1]. By monitoring household energy consumption, HEMSs can provide consumers and the utility with energy-saving opportunities through control, scheduling, feedback, and recommendations [2]. Accordingly, this service is an essential prerequisite of HEMSs to observe electrical load and actuate energy management strategies. Load monitoring can track energy use at different electrical layers comprising household level (main entry), circuit level (electrical panel), plug level, and/or appliance level [3]. From the viewpoint of potential applications, information about energy usage behavior at the most disaggregate level, i.e., the bottom layer, is preferred. From a technical point of view, exploiting only the aggregate data measured at the entry point, i.e., the top layer, is suggested [4]. These two standpoints stimulate the non-intrusive approach to load monitoring that identifies operating appliances and estimates their individual energy consumption in the aggregate signal. Non-Intrusive Load Monitoring (NILM) emerges as an alternative to troublesome intrusive practices (ILM) depending on sensing systems and sub-meter measurements located after the main entry [5]. It can provide viable assistance to HEMSs. NILM can characterize customers based on their appliance usage patterns, describe them according to their tendency towards utilizing their devices, and identify those with a higher propensity for participating in power system operation [6]. Consequently, it can facilitate recognizing the topmost end-users for energy efficiency services, Demand Response (DR) programs, and electrification. Research studies show that NILM can result in a significant energy saving of up to 15% by giving detailed energy feedback to consumers [7]. Figure 1 explains intrusive and non-intrusive means to execute load monitoring at different layers of household electric circuits [8].

### Motivation

From a feasible viewpoint, smart meters are the modern metering infrastructures to enable cost-effective energy-saving solutions through NILM in the residential sector [1]. These emergent technologies stimulate NILM operations adopted for data acquisition systems with a low sampling rate due to their technical limitations [9]. In this context, only steady-state characteristics of power demand can be exploited for load disaggregation. These features mostly belong to active power since supplying other readings increases the price of smart meters. For example, monitoring reactive power, in addition, can raise expenses by 50% [10]. The literature has a similar trend in providing the information space, thus supporting a realistic concept. However, it draws on data with fine granularity induced by the electrical properties of public databases. The sampling frequency of regular datasets on which relevant research relies can be acknowledged as high for actual measurement apparatus [11]. Regardless of the mathematical model, the most common databases for NILM practices contain measurements with a sampling rate of ≥1 Hz [12]. This fact is confirmed by the four most cited datasets, i.e., REDD, UMass Smart, UK-DALE, and BLUED [13]. Furthermore, among the ten most cited ones, only two have a sampling time of ≥1 min, i.e., AMPDs and HES, as shown in Table 1 [1,11]. The public body of information can also influence other NILM exercises. For example, a technical reason for suggesting event-less methods can be that most databases do not provide a means for the analysis of the event detection phase [14]. On the other side, the NILM evaluation framework, specifically Deep Learning (DL)-based analyses as the state-of-the-art, has mainly focused on the energy estimation and load reconstruction of fridges, kettles, microwaves, dishwashers, and washing machines as the targeted loads [15,16,17,18,19]. Among these appliances, only the last two have sensible flexibility potentials considering their controllability and usage, regardless of their identification precision, as another challenge [20]. The first three devices are essentially uncontrollable since their manipulation can jeopardize customers’ activities. In addition, the set of intended loads neglect equipment like Air Conditioning (AC) systems and Electrical Vehicles (EVs) with a high degree of control that can bring about practicable grid services, particularly DR [21]. This inadequacy can also be attributed to the nature of public data. Such circumstances narrow NILM’s potential for useful applications.

The issues above can cause insufficiency in contemporary NILM techniques for practical applications. They can question the generalization, effectiveness, and dependability of case studies to make NILM technically feasible. The concern is raised when relevant research looks for solutions in geographically-specific regions with unexplored energy consumption cases [22]. This situation matters to the Quebec area, where the type of household electrical appliances composes an energy demand profile that cannot be typified by existing public databases. Accordingly, applicable data become imperative to investigating NILM scenarios created by residential load behavior in this region, particularly based on low-sampling disaggregation methods. Such information can bring about an opportunity for analyzing the high flexibility potentials of Quebec household energy usage manifested by loads that rarely exist in public datasets. These aspects signify geographically-related NILM analyses based on data from smart meters. Such examinations can assist in observing actual barriers to achieving a fruitful NILM system. Accordingly, this paper tackles the problem of low-sampling-rate NILM in the context of Quebec. Its realistic case study utilizes on-site smart meter readings of the active power of a set of houses with a sampling interval of 15 min. In order to provide a clear picture of the specificity of the case study, it commences with a comparative statistical analysis by exploiting data from European regions. Afterward, the study proposes a load identification mechanism that takes advantage of smart sub-metering technologies and offers a combination of ILM and NILM. The proposed approach aims to relieve the complications related to a system solely based on NILM in Quebec residences. In accordance with the recommended solution, a load disaggregation method is also suggested on the basis of supervised and unsupervised machine learning techniques to demonstrate these difficulties. In addition, this work elaborates on other practical solutions suggested worldwide to enable a feasibility study through its algorithmic design. As a result, it launches applied research toward practical NILM in Quebec and other districts with similar circumstances. The rest of the paper is organized as follows. Section 2 briefly discusses basic research on NILM. Section 3 represents the statistical analysis aimed at comparing household energy usage properties of Quebec data with public databases. Section 4 explains the proposed method fortified by the suggested NILM practice in Section 5. Section 6 presents the results and discussion, followed by conclusions in Section 7.

## 2. Fundamentals of NILM Concept

NILM, proposed by Hart in 1992, has been practiced for many years based on load disaggregation algorithms to separate the household electrical load into a set of individual appliances [23]. The common facet of these methods comprises feature selection, classification, and recognition through exploiting data collected by an acquisition system [1]. Figure 2 presents these steps along with their general analytical processes, which outline the disaggregation engine. Different aspects considered to configure these procedures are briefly discussed with relevant references for further exploration.

### 2.1. Mathematical Methods

Numerical schemes used to formulate an NILM task can be mainly classified into event-based and event-less algorithms. Unlike the latter, which inspects all data instances, the former is more computationally efficient, since it draws inferences only about detected events. On the other hand, the second can rectify incorrect estimations since it evaluates every sample for classification dissimilar to the first, which can contain inaccuracies due to misdetection or false-detection [24]. Additionally, the mathematical approaches to NILM can be categorized into optimization, traditional machine learning, and DL models. Variants of Hidden Markov Models (HMMs) are the most famous former NILM method that can provide an efficient representation of appliances [6]. Linear and non-linear mixed integer optimization are other popular means of load identification in the literature [25,26]. Recently, DL, mostly Recurrent Neural Networks (RNNs) and Convolutional Neural Networks (CNN), has drawn significant attention for performing NILM [27]. Mathematical techniques for load disaggregation can be fundamentally exercised through Open Source NILM Toolkit (NILMTK). NILMTK provides a public framework for executing NILM based on different methods over several public databases [28].

### 2.2. Learning Procedures

An NILM system established by machine learning methods can be divided according to its learning phase. In this regard, it is developed based on supervised, semi-supervised, and unsupervised means. Supervised techniques, as the most common strategy, use labeled data to train the disaggregation model. Semi-supervised methods use partially labeled data or a set of prior knowledge for the same purpose. Unsupervised practices otherwise consider no labeled data for training [29]. The operating systems based on the first learning concept form a supervised NILM, while those based on the two latter training mechanisms create an unsupervised NILM exercise [30]. Supervised NILM requires relatively more human intervention and computational resources. It is more promising at the cost of sufficient data, where it attempts to identify a wide range of appliances. Such a system uses static information to create load models, thus is indifferent to variations in appliances’ temporal behavior and the presence of unknown devices in the total signal (see Figure 2). Unsupervised NILM demands manual labeling. However, it is dynamic regarding temporal changes in the electrical signal since its identification task is independent of targeting a specific set of appliances through their models. This operation can assist with the low-cost generalization of accurate algorithms appealing to business [6].

### 2.3. Operation Modes

The operating mode is another important aspect of an NILM system. Generally, a real-time operation is favored to enable practical applications for HEMS services. A low-complex highly-accurate algorithm is essential to operate a real-time disaggregation. All research that intends real-world implementations should adopt such a manner. In this regard, the learning phase of an NILM exercise can be carried out through either online or offline means according to its algorithmic process, as shown in Figure 3. Off-line training accounts for a manual configuration that uses sub-measurement readings while online learning presents an automatic configuration that utilizes only general information (see Figure 2). Supervised NILM methods are basically offline processes with two separate steps of model learning and load identification. On the other hand, unsupervised NILM schemes promote online procedures with adaptive structures since they utilize no/less training [5]. From the perspective of specific applications, both frameworks can offer a real-time service if they guarantee pre-defined deadlines [31]. Enabling the real-time aspect is a great deal for any load monitoring system since delayed/late usage information is restricted to empower customers for near real-time load modification. For example, major loads, such as HVAC systems, are influenced by immediate end-users’ demand and, thus, modifying their current patterns requires on-time actions [6]. It is important to know that a level of prior knowledge is the least requirement of any effective NILM. This, a priori, can simply be the number of states, the power levels, or the operation time of appliances so that their electrical characteristics are not subject to large variations across households [5]. Primary information can be acquired from manufacturers or experts. Regardless of the operation mode, a totally blind NILM process can lose crucial features and fail in identifying loads accurately [6].

### 2.4. Data Source

The development of a fruitful NILM vastly depends on the quality of data, principally its sampling intervals. Data frequency plays a key role in the algorithmic design and technical configuration of an NILM system, discussed above [22]. High-frequency data are collected at either thousands or millions of times per second using specific hardware. Lower sampling rates of 1–10 per second and 15–30 per minute are extracted from smart meters over a Home Area Network (HAN) and a Wide Area Network (WAN), respectively [32]. For high sampling rate data, methods that can deal with harmonics and transient (microscopic) features have been explored. These techniques require excessive learning and face difficulties in measuring energy consumption and realizing robust procedures [33]. Contrarily, algorithms that can handle power features, such as being active and reactive, have been investigated for lower data granularity. Such means are generally applied to the steady-state (macroscopic) electrical characteristics of appliances’ operating states in the context of a power-based load disaggregation [34]. Such cost-effective NILM systems process electric power signals that are easier to measure and transmit [35]. However, steady-state-based mechanisms can undergo challenges related to spatial and time overlapping of appliances operation, specifically those with lower power amplitude and higher occurrence [36].

In addition, NILM can be assisted by high-level macroscopic properties of appliances’ operation habits as secondary information. For instance, Time of Use (ToU) is an additional feature that can be utilized for identifying loads with certain operation times more accurately [37]. Nevertheless, different concerns should be noticed while using secondary information. The enhancement of appliances’ load recognition should be worth the cost of processing their additional characteristics. The improvement in energy estimation should not hinder other applications. For example, utilizing ToU information should not avoid uncovering temporal abnormality intended by anomaly detection services. Data required for integrating secondary knowledge into the training phase of the disaggregation method should be accessible. For instance, capturing the ToU characteristics of devices with a lower occurrence, like a washing machine, needs a considerable amount of data; this is a crucial issue since NILM already suffers from reliable data limitations. Notwithstanding, secondary information, such as time of day, duration of usage, and correlation effects, can be used to deal with challenging disaggregation practices [3].

## 3. Quebec Residential Energy Usage Context

The difficulty of NILM in Quebec can be ascribed to load specifications that lead to a disaggregation scenario with multiple complexities [38]. NILM complication is mainly rooted in the number of appliances, their power level differences, and the frequency of their state of operation changes, which are all exposed by the case study. Except for common loads, Quebec dwellings are mostly equipped with electric space and water heating systems due to particular geographical conditions. Each house is equipped with several Electric Baseboard Heaters (EBHs) (8–12 numbers) with high switching frequencies. These loads can have power levels that are similar, not only to each other (almost identical for the same products) but also to other energy-extensive appliances like Electric Water Heaters (EWHs), washing machines, and dryers. Additionally, EBHs can distort low-power trajectories associated with a wide range of devices, such as fridges, freezers, lighting, entertainment equipment, and even kettles and microwaves, due to their high power usage and lengthy operation time. This set includes more than half of the appliances (three out of five) targeted by load disaggregation in the basic research, as mentioned previously. EWHs deteriorate this situation with their large demand, short duration, and regular presence. These loads share similar issues with EBHs, like challenging other loads’ operation detection [22]. EWHs are also a dominant load in other geographical locations. For instance, in New Zealand households, water heating presents the highest demand with 27% and, thus, is considered a crucial element of demand-side management strategies [39]. The aggregate load profiled under such conditions yields a monitoring scenario rarely experienced by current disaggregation algorithms. As described, the main reason for this negligence is the focus of the methods on public databases, hardly exemplifying cases like Quebec. In [40], the authors investigate the impact of EBHs on aggregate power consumption by adding the overall demand of only four baseboards to a daily load profile from the ECO dataset. For further analysis of the Quebec context, smart meter data from ten residences with 15 min sampling intervals are utilized. For these houses, aggregate and circuit-level power consumption at a 1 min sampling rate is also available for an elaborated investigation, which is not a real-world condition.

### 3.1. Quebec Residential Data Features

The primary challenge with developing an NILM system for Quebec houses is the diversity in energy statistics across seasons. This situation is observed by exploiting 1 min measurement data from different houses, numbered for ease of understanding. Figure 4 exemplifies the behavior within the warm seasons. The first aspect of the aggregate power profile measured through the ’main’ input of the panel is high fluctuations, especially in lower demands. In all instances, this can be mainly attributed to the tedious time-varying load of the Heat Pump (HP). In combination with lower demands from appliances like the fridge, this behavior creates a challenging pattern to disaggregate (Houses 1 and 2). The second feature of the overall use in summer is the similarity of the power usage between energy-expensive devices comprising EWH, dryers, spas (House 3), and even stoves (House 2). This characteristic can increase the overlapping rate, mostly with EWH due to its regular operation, sometimes with two different levels of demand (House 2). Furthermore, the consistent occurrence of the pool (House 4) can expand the concern with this region of demand. Indeed, overlapping among major appliances is hardly experienced in public databases, knowing that washing machines and dishwashers are not measured in the case study, which can boost the problem. Notwithstanding, the case of Quebec demonstrates its specificity by presenting the load of several EBHs. Figure 5 exemplifies this condition for two houses with different numbers of baseboards. Each circuit-based reading, the Heat Channel (HC), represents two heating devices that share the Total Heat (TH). There is no doubt that such usage can have a massive impact on the domestic load. This influence can be noticed through two visible features comprising dramatic variations and significant demand compared to the overall domestic load (see Figure 4), which can dim their patterns. It forms a ’main’ power consumption as the aggregate usage in cold seasons that is thoroughly different than in warm periods. For example, the effective fluctuations of HP, sometimes bigger in winter, are not observable under the new situation.

### 3.2. Quebec Comparative Data Statistics

The careful observation of power consumption patterns in the Quebec house samples can be elaborated by a comparative analysis based on public databases. For this purpose, UK-DALE and ECO are exploited, which belong to the ten most cited datasets. The former has been broadly utilized in recent studies based on DL and the latter provides challenging cases for an NILM task [5,15]. In addition, these popular datasets hold data measurements for an extended period of time that can reveal power consumption patterns impacted by weather conditions and customers’ activities. This feature, as the reason for excluding REDD, is essential for a sensible comparison with the Quebec data, since its behavior widely varies in accordance with seasonal changes. Insufficient measurement is also the logic behind omitting House 3 in UK-DALE from the analysis. The statistical exploration is extended by using other data from a load monitoring study funded by the CleverGuard (CG) project (“CleverGuard—ICT solution based on energy data for protecting the elderly at home, staying safe and independently” an EU funded AALJP project, 2021–2023, contact: Pascal Kienast, pascal@clemap.ch) in Switzerland [41]. This confidential information was obtained from a set of customers and was recently shared with the authors under collaborative research work. Although CG data are private, they are referred to as public for simplification. For the Quebec case, the statistics are explored for both aggregate and group-level data comprising domestic and TH loads. Subsequently, the following statistics are applied to four selected databases to enable an effective comparison.

Figure 6 depicts the public and Quebec data distribution in corresponding houses at a 15 min sampling rate to manage its size. The terms in this figure represent data name and house number; for example, QH2 stands for Quebec House 2. Generally, the level of power demands is not comparable between both cases in any region of scattered data from all instances, even with the focus only on the Quebec domestic load, which can share similar appliances with the public databases. With regard to the interquartile and whisker range, representing half and all data, respectively, it can be deduced that EBHs, EWHs, and case-specific devices remarkably change the spread of demand in Quebec houses.

With reference to the outlier ranges in public data, it is more likely that this region includes samples related to energy-extensive loads due to their power levels and operation schedules, except for ECO Houses 4 and 5 (ECOH4 and ECOH5) and CG House 4 (CGH4). Therefore, it can be stated that, in most cases, major appliances operate in distinguishable regions of public load profiles since outliers are data points with significant differences from the rest of the samples. Such a circumstance facilitates identifying these types of devices, such as washing machines, dishwashers, and kettles, as targeted loads in the NILM literature. Indeed, the outlier extent in Quebec data distinguishes no appliances, either targeted or non-targeted.

Figure 7 indicates the frequency histograms of public and Quebec data.

For the former, a 1 min sampling period has been used to better approximate the active power of the operation state of existing appliances and to provide insights into probable groups of targeted ones. It can be observed that a substantial portion of samples has a power demand of less than 500 W in the public data. Such similarities can be challenging for NILM if it represents the demands of several targeted devices. However, the only major operation in this power band relates to the fridge. This can be noted by investigating these databases at the appliance level and reducing power intervals in the analysis. For example, close to 50% of samples carry a load of less than 200 W in around 70% of cases. Additionally, a minor fraction of instances lies over 1 kW which, interestingly, contains power values of other targeted appliances. Particularly, the washing machine, dishwasher, and kettle operate in this boundary according to appliance-level information of associated datasets. Knowing the fact that these loads advertise an operation schedule, such specificity can assist with their load identification. Indeed, in all cases, a cluster with power quantities over 1300 W can be approximated, which stands out of 90% of all the data. On the other hand, none of the above distinctive patterns is manifested by the frequency histogram of Quebec data, specifically regarding the appliance target space. In this case, only 50% of samples cover a wide range of up to 3 kW split into several groups with significant frequencies.

The power consumption pattern is another characteristic that can provide sensible insights into the data. Exploring this property can help improve demand-side management strategies by understanding customers’ behavior toward utilizing their electrical appliances, especially based on activity cycles and climate conditions [42]. Although pattern recognition should be an essential service of any load monitoring system regarding energy-saving awareness, popular databases are inadequate to enable such a practice due to limited data length and quality. This can be observed in Figure 8, where it is challenging to determine a common period to draw inferences about behavioral differences among end-users. Notwithstanding three years of data acquisition, the UK-DALE database seriously suffers from missing data, and the available data are scattered across dissimilar time periods, except for House 1. However, a continuous pattern has been extracted by combining relevant houses for six months. House 3 does not offer sufficient readings, even for an individual analysis. The ECO dataset is subject to the same problem with less severity since, except for Houses 3 and 6 with notable missing data, power values are available for roughly the entire measurement period. Nonetheless, it can be noticed that diurnal behavior per month is similar for each case study in the public data. Slightly higher variations can be detected within time progress in the ECO data.

Since the selected period covers partially warm and cold seasons, such a similar occurrence demonstrates less correlation with environmental factors and a stronger relationship with calendar ones. The pattern tendency can be attributed to both the type of in-use appliances, especially heating/cooling systems, and the climate condition. These are the same reasons for which the Quebec data illustrates a significantly different usage pattern, as shown in Figure 9a. Weather and calendar components strongly influence the power consumption behavior in Quebec houses. A major share of this impact can be assigned to the notable load of EBHs, according to Figure 9b [43]. It should be noted that a total number of eight houses have been exploited for this statistic due to the lack of measurement data at the main circuit for two houses.

Exploring power demand time series according to their systematic and unsystematic components is another useful analysis regarding data characterization. A seasonality study can be employed for this purpose. Figure 10 exemplifies this exercise for two cases. From public databases, ECO is considered regarding its demand pattern, and House 4 is selected considering its power distribution, which signifies the presence of energy-demanding loads, especially seasonal ones. The examination is carried out by the use of the multiplicative model since it is a better choice for time-varying behavior and removes difficulty in interpreting negative values. Furthermore, this proves to perform better in capturing peaks through its seasonal component. With regard to the residual element, it can be noticed that the public case contains a huge amount of unpredictable/noisy information, which exposes fluctuations almost similar to the main signal. This shows that a notable amount of data is not consistent with the rest of it. The trend and seasonal factors show a general upward slope and a clear recurring/periodic pattern, respectively, that are relatively poor considering the residual. This can be estimated by multiplying these two components. It is observed that the systematic information contributes inadequately to explaining the usage. On the other hand, the Quebec data are characterized by valuable systematic information, particularly in winter, where the model is able to strongly describe the demand. In addition, it can be realized that TH has a great influence on the seasonality of the aggregate load. This impact, along with the level of overall systematic information, promotes a seasonality-based NILM approach to disaggregating EBHs load. In addition, the results demonstrate that a classic decomposition is not an efficient choice for the Quebec case, since the seasonality of the data strongly changes within the year. It should be noted that the seasonality of other public instances is inferior to the selected one, specifically for UK-DALE and CG data.

In order to discover the real-world relationship among data instances and suggest generalizable hypotheses, a correlation analysis can be used. Figure 11 presents the results of this investigation into existing houses from all datasets, except for UK-DALE due to dissimilarity over available data periods. It can be seen that the correlation between household usage in the public data is not even moderate, evidencing a notable difference in their tendency toward operating electrical appliances. Similar behavior can be detected in the Quebec domestic load. However, a moderate correlation can be noticed across the overall load that is rooted in the medium to high correspondence between TH demand, similar to seasonality.

The above analysis has been aimed at revealing statistics that have not been fairly taken into consideration in related research and can be applied to any other case studies. Indeed, it is evident that the Quebec data encounter a vastly higher amount of events compared to public data due to a bigger number of appliances with high switching frequency in relevant operations, i.e., EBHs. From every targeted aspect, the statistical analysis demonstrates massive differences between the public and Quebec data. From a practical standpoint, a load monitoring practice should approach this case with a different set of targeted appliances. The group of interest must certainly contain EBHs due to their share of demand, their impact on load characteristics, and their potential applications for HEMSs. It should also include EWHs due to their major usage and regular presence that makes them responsible for the most rapid rises in household demand all across the year. It can also be acknowledged from the statistical study that disaggregating the choice of appliances in the literature, especially fridges, kettles, and microwaves, from the Quebec data is a burdensome task. This exercise becomes more difficult knowing the fact that actual readings have a sampling rate of 15 min. Figure 12 shows daily household load profiles recorded by smart meters within warm and cold seasons. Complex operational curves, seasonal variations, and continuous changes at lower demands are the challenging features of these profiles. In summer, the EWH represents the most notable demand, contributing to almost all load peaks solely or partially. As a result, it gives the total usage a similar pattern to its own in both shape and magnitude. In winter, the EBHs illustrate their remarkable influence by transforming the domestic load as shown in this figure. Another important property that can be realized is the significant level of unknown demand as the difference between the main circuit and the total loads and domestic and aggregate usages in Figure 10a,b, respectively. This underlying issue can seriously impact the performance of an NILM practice. Accordingly, it can be stated that NILM faces a completely different case in Quebec dwellings.

## 4. A Disaggregation Approach to Quebec Household Power Consumption

As discussed, the designation of an NILM model highly relies upon data sampling intervals. The least requirements reported for a disaggregation task based on regression and classification methods are 3 Hz and 1 Hz, respectively, which is not the case in this study [11]. From the viewpoint of feature extraction, power-based steady-state electrical features at a macroscopic level are recommended for load disaggregation purposes. Nevertheless, relevant research indicates that even a sampling time of 1 min is not sufficient to capture EBHs’ behavior due to their rapid state transitions. From the load identification point of view, an unsupervised method is favored for the Quebec case due to data shortage in both length and quality, e.g., appliance-level information. Such an approach can be combined with a supervised technique to relieve the impracticality of a completely blind practice at the cost of sufficient data, particularly for EBHs with seasonal presence. These aspects help outline an NILM system for the Quebec case. Notwithstanding, general inadequacies of existing methods in the literature on one side, and potential challenges of deploying NILM in the Quebec region on the other side, stimulate alternative solutions.

As illustrated in Figure 1, ILM is another technology that can take part in HEMS strategies. This method of load monitoring can accurately determine the operation state of a device with the use of one or several sensors. Although the barriers related to complex installation and multiple sensor configurations can limit this approach, affordable technology and reliable implementation can support it. The ILM can have different levels according to the number of meters, which can be one meter per zone, plug, or appliance. The former has the lowest complexity in terms of both hardware and software as well as data transition requirements. Therefore, it can be simply combined with the entry point meter (non-intrusive concept) to boost monitoring efficiency, particularly in small-scale installations, for example, only for major appliances [10,38]. Similarly, in Quebec households, there is one thermostat per zone for controlling corresponding EBHs. The information on the thermostats can be combined with that on total energy demand to improve NILM results. HEMS can communicate with each EBH (if all thermostats are connected) or zone (otherwise) and use the acquired data to disaggregate the overall heating use from the aggregate load profile. Subsequently, the remaining usage can be analyzed to uncover the power demand of EWHs as the second targeted load in Quebec households. Extracting these two demanding loads allows for the exploration of the residual to disaggregate common home appliances, i.e., those from the literature, based on popular NILM methods. This potential solution draws interest to the approach of combining ILM and NILM systems in Quebec houses for alleviating its complicated scenario. This method is also supported by advancements in IoT technologies that link smart appliances throughout the house. Figure 13 illustrates the proposed approach, where EBHs data can also be obtained from the electric panel to extract their total demand. In this illustration, a piece of prior knowledge acquired from either the same dwelling (1) or other houses (2) can assist the whole procedure.

Avoiding the disaggregation of devices at one single point is a feasible approach that can increase accuracy, since with more than one accumulating point, more classifications can be executed. In the Quebec case, one point can disaggregate total heating demand into individual EBHs and another can disassociate EWH and other loads. Indeed, a practical load monitoring structure should focus on both development cost and proper accuracy. A combinatory method can realize these two goals with an effective number of meters, simpler signatures, straightforward algorithms, and less complicated accumulating profiles. The feasibility of this approach can be fortified by applicable NILM products like Bidgely, which use programmable thermostat data to improve the performance of disaggregation techniques and offer optimal setpoint schedules. It is also supported by the Bidgely low-resolution disaggregation technique. The relevant product uses the low-granularity whole-house profile, non-electrical information, and training data to identify individual appliances’ load profiles. As stated by Bidgely, a reliable NILM on low-resolution data should be carried out by reasonably specific underlying techniques. One method can extract a portion of energy from overall usage related to a specific class of loads such as pool pumps, air conditioning, furnaces, and EBHs in our case, while another scheme can use a training phase based on specific signatures/parameters to disaggregate power consumption patterns of other categories of appliances [44].

## 5. An Introductory NILM Practice in Quebec Residences

Although the proposed approach implies a mechanism that takes advantage of ILM to extract EBHs demand from the aggregate usage, it is practiced by a procedure entirely based on NILM to demonstrate load disaggregation complications in Quebec houses by numerical outcomes. The methodology follows the steps indicated in Figure 13. As shown in Figure 14, the initial task focuses on extracting TH demand from the aggregate load profile through a supervised method, while the successive one concentrates on capturing other possible devices in the remaining signal within an unsupervised process. In addition to facilities brought about by thermostats, the statistical analysis also proves that heating load should be the first target in Quebec houses.

The energy estimation of electric heating systems has also drawn attention to other implementations. In this regard, analyzing heating systems’ relationships with external factors and time-series patterns have been favored. In France, the Hello Watt Company has carried out the heating demand disaggregation according to its relationship with outdoor temperature by means of the piece-wise linear regression method [45]. However, such techniques cannot efficiently explain the heating load in Quebec residences due to its complex behavior. Our studies on Quebec data show that individual household demands and outside temperature are not sufficiently correlated to be treated by linear designs [43]. To be specific, heating demand is significantly influenced by calendar components related to occupants’ activities as well. Therefore, meteorological variables are inadequate for accurately estimating this consumption for load identification purposes. In Switzerland, a similar examination has been performed by means of time-series clustering methods [46]. Nevertheless, the utilized scheme is not applicable to the Quebec case since it is aimed at recognizing a water-based heating system with a large distinct demand of 18 kW occurring at periods with the lowest power consumption, i.e., between 23:00 and 6:00. As shown in Figure 2 of the related study, the pattern of an electrical load with these properties is not ambiguous and is thus difficult to extract.

As a state-of-the-art approach to NILM, DL is another promising method for identifying the total demand of EBHs and disaggregating it from the aggregate load profile. Particularly, Long Short-Term Memory (LSTM) networks can be employed for this task due to their capability to learn long-term temporal dependencies in time-series. Accordingly, an LSTM network is considered to execute the supervised phase of the suggested methodology. The structure is developed in Python by using Keras as a DL API. In this regard, a stacked LSTM architecture through a sequential model is constructed. The selected arrangement has five hidden layers and one output layer. The hidden architecture contains three fully connected LSTMs with 30, 30, and 20 neurons, and two Dense layers with 40 and 20 neurons, progressively. Adam and mean square error in terms of optimization and loss functions, respectively, are used to compile the network. In addition, tanh and relu are utilized as non-linear and rectified-linear activation functions in the first and last layers, respectively. Training is carried out within 100 epochs running a sliding window that maps seven samples of aggregate power demand to the next (eighth) one of the overall heating load. In addition, its convergence is enhanced by utilizing a dynamic learning rate.
Figure 14The block diagram of the NILM practice proposed to tackle the Quebec case [47].
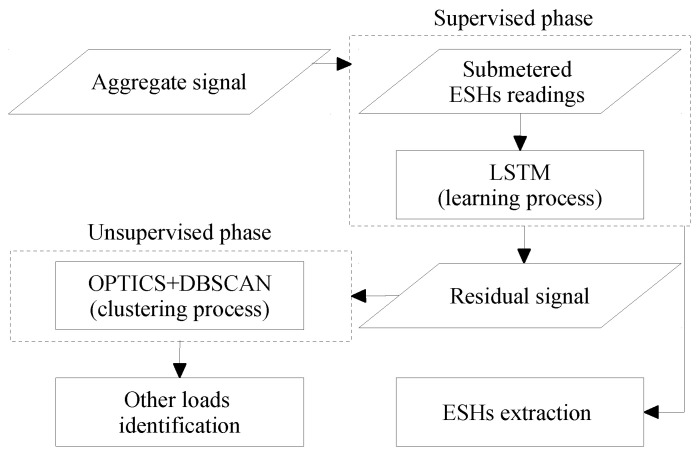



After subtracting the EBHs energy use, the remaining signal is analyzed for identifying other possible appliances in an unsupervised context. This framework takes advantage of a clustering scheme that focuses on grouping similar demands according to a probability threshold, determining their occurring time in the power profile, and creating their related operation (ON-OFF) sequence, which can possibly represent a load. The on-state of each created sequence is defined by calculating the average values of the corresponding cluster. This process can be carried out by using either power level or power difference information [5]. Particularly in an unsupervised framework, the granularity of data has a significant impact on the soundness of sequences, captured across the time-series data. Figure 15 illustrates the power level and difference information of EWH in House 1 at one and 15 min sampling intervals. The most evident statistic that can be acknowledged in this figure is the critical change in the power demand pattern at the lower frequency. An unsupervised system can observe this trajectory as a combination of several loads. In a supervised practice, such behavior can cause different loads to be represented with similar models. Therefore, in both cases, recognizing a load with no prior knowledge becomes a burdensome task, and exploiting expert/secondary information is necessitated. Moreover, it can be noticed that the sampling rate can notably alter the ON-OFF power transitions of EWH. They are quite different even in the higher sampling frequency, knowing the fact that this resistive appliance has one of the most uniform steady-state behaviors among household devices [22]. In this situation, utilizing power difference for sequence construction can result in incorrect state matching. It can also lose high operation states and, in turn, neglect an important share of demand. As a result, employing power level is preferred since it creates sensible operation sequences of potential devices despite its variations and overlapping events. All appliances can be subject to this transformation under a low sampling rate measurement, especially those with short operation duration.

Two different techniques have been evaluated for executing the clustering phase. The first method employs a Kernel-based Subtractive Clustering (K-SC) algorithm [29]. The second scheme takes advantage of DBSCAN (Density-Based Spatial Clustering of Applications with Noise), which enables time-series clustering. Accordingly, its advantage over the K-SC method is that it executes the first two steps of the sequence creation process (load classification and occurrence detection) within one stage. Nevertheless, DBSCAN requires the definition of two parameters that account for Epsilon and MinPts. These parameters explain that the Epsilon-neighborhood of each point must contain a minimum number of points (MinPts). A specific way to estimate MinPts has not been suggested in the literature. For example, the authors of [48] have recommended using a value twice the dimension of the data. This suggestion is adopted for this analysis and, thus, a MinPts of 4 is selected. Additionally, particular methods have been developed to compute Epsilon. In [49], the authors have provided a procedure within which, first, the average distance between each point and its k (equal to MinPts defined previously) nearest neighbors is calculated, second, k-distances are plotted in ascending order, and third, the biggest slope (greatest curvature) is considered as an Epsilon value. This process has been evaluated for the case study in order to provide an effective application of DBSCAN. Figure 16 exemplifies the outcomes for the power level and difference data from House 1 on a typical day. It can be noticed that the method is not effective for the power difference data since the distance plot fails to realize an elbow at which a specific point can be chosen as the Epsilon value. On the other side, the elbow plot of the power level data yields a sharp slope. Nonetheless, this curvature covers a wide range of points corresponding to samples from 70 to higher than 250, all of which can be used for the Epsilon parameter. The analysis evidences that moving across this range causes DBSCAN to create different clusters. This, in turn, makes the choice of Epsilon a difficult task, especially in an unsupervised system. It should be noted that this challenge is not necessarily the case for other types of data and can be attributed to Quebec data characteristics (see Section 3.2). However, it necessitates exploring other schemes. In this regard, OPTICS (Ordering Points To Identify the Clustering Structure) is considered as another density-based clustering method. OPTICS is able to provide information about cluster structure in the time-series data that can be observed through the reachability plot. This plot is illustrated in Figure 17 for the same MinPts and instance as in Figure 16. It presents the structure of the clusters in the data according to the peaks and valleys. For instance, in Figure 17a, three major groups can be collected considering the significant peaks. In fact, the interpretation of the significance can define the number of clusters as noticed in this figure. It can be observed that OPTICS not only facilitates the choice of Epsilon but also defines the prospective clusters. This technique can manifest the change in the number of clusters based on the range of Epsilon values. This can be simply practiced by drawing horizontal lines on the reachability plots in Figure 17. As a result, a combination of the OPTICS and DBSCAN algorithms is suggested for the sequence creation step. Generally, the closest values to the top of the significant peak with the least amount are considered the best candidates for Epsilon. This value is set to 0.4 in all cases. Different analyses demonstrate that the combined clustering approach proposed and the K-SC method generally capture comparable operations. However, the former uncovers a higher number of sequences for a similar cluster, which can reflect a higher estimation of energy demand for a possible load. Moreover, DBSCAN provides a class of outliers belonging to no cluster that can be further analyzed.

## 6. Results and Discussion

The developed NILM system is applied to the nQuebec data. For the supervised phase, all ten houses are evaluated. The developed LSTM model is trained for each case by exploiting EBHs data. The main concern over this network, like the other DL structures, is the need for sufficient data. This matters in all case studies since the data span only one year. The amount of useful data reduces knowing the fact that the abundant patterns of EBHs for learning and testing are mostly available during cold seasons. In order to alleviate this problem, 90% of the data are considered for training, which can cover more winter samples. For the quantification of the outcomes, the mean absolute error (MAE), the mean square error (MSE), the root mean square error (RMSE), and the symmetric mean absolute percentage error (sMAPE) are exploited to provide a thorough examination [50]. Table 2 presents the test results of the Quebec houses. The MAE should be analyzed comparatively. For instance, the heating demand reaches up to 18kW in House 2, which causes a high error. Additionally, the percentage score improves the interpretation of the outcomes. The values of sMAPE along with MAE demonstrate that the model competence in TH demand estimation is acceptable in Houses 1, 5, and 9. The difficulty in capturing load peaks entirely is the main source of inaccuracy in these three houses. Nevertheless, their LSTM design outperforms the others regarding MSE and RMSE rates. In addition, the LSTM performs fairly in Houses 3, 6, and 8 considering the large power levels of EBHs usage. On the other hand, the results in Houses 4, 7, and 10 are poor. Generally, the DL model’s inadequacy in estimating power spikes can be improved by analyzing the source of such behavior. The study shows that these patterns can be related to specific activities due to their periodicity. Moreover, the network scheme can be enhanced by adding other sources of information, e.g., descriptive components of power demand since the ongoing practice relies upon only one input. Such variables can be simply supplied to the HEMS and used in different operations. It should be added that several LSTM networks with different structures, data scalers, demand types (energy), and input shapes have been exercised among which the current model provides better results. Notwithstanding the limited amount of data, the share of EBHs’ demand in overall usage, and the heating trend pattern, its sudden fluctuations with large values are critical issues that should be dealt with in any intended structure.

Furthermore, Table 3 lists the results for House 1 over one week in winter based on MAE, MSE, and the Proportion of Total Energy Correctly Assigned (TECA) metrics [50]. With regard to TECA values, it can be deduced that the regression model performs EBHs load profiling adequately knowing that a satisfying load disaggregation task should yield a minimum accuracy of 80% [5]. From the results, it can be stated that the method carries out the supervised step efficiently since the TH used in House 1 has a high value of 10 kW. Nevertheless, the assessment error results in two inevitable issues that are passed along to the next step accounting for inefficiency in catching peaks and variations in estimating lower demand. Considering the fluctuations in the overall demand, the second matter has a higher impact on the remaining load. The situation worsens in the presence of unknown demand. In fact, the share of this type of usage varies from 16% to 49% in targeted houses (26% in House 1), which is remarkable regarding the power consumption ranges. The adverse impact of these circumstances on the unsupervised phase is creating impractical clusters with wide variances in the corresponding region of the load profile. This is acknowledged by applying the proposed clustering technique to residual demand across the same week of Table 3 to uncover valid load operations, as shown in Table 4. It is evident that the first loads represent no actual devices. This is the effect of the error in EBHs load estimation that makes the recognition of devices with low power use doubtful. This condition can be the case for any system with a similar strategy since the power of small loads, such as refrigerators, is normally close to the inaccuracy of examining the demand of energy-expensive devices like EBHs.

Accordingly, the four last loads are considered for appliance recognition. The worst outcome occurs on the third day when the sequence creation is not able to identify any possible load. Nonetheless, the outcomes for such days can be improved since a fixed Epsilon value is intended for all the days. Besides, the weighted clustering demonstrates the existence of a pattern in the occurrence of specific loads, for example, ≈2.9 kW and ≈1.9 kw in the second and fourth categories, respectively, which increases their feasibility as actual appliances. In a 15 min interval, the clustering outcomes reveal the concerns, discussed above, about load identification in the residual signal. First, the notable loads become very similar in such a way that their separation is almost impossible within an unsupervised system. This causes a created sequence to describe more than one device. Second, the behavior of an appliance changes within consecutive intervals and, thus, the sequence construction either explains its operation by different loads or fails to capture it completely; third, the rate of overlapping increases. These situations cause a complicated interpretation of valid loads in Table 4. It can be noticed that, on the 4th day, one possible load stands for three actual appliances, while on the 1st day, one actual device is captured through four possible loads. As expected from the statistical analysis, all the loads in Table 4 are related to EWH since it holds the highest portion of overall usage. In fact, EWH either represents a load or shares it with other devices. It should be noted that, in some cases, this share is very delicate. For instance, on the 4th day, only two sub-sequences of the relevant created operation are associated with the stove and dryer. On the 7th day, only one sub-sequence corresponds to the stove. However, the overlapping of the biggest usages, like the high demands of EWH and dryers, is rarely faced since the developed clustering procedure considers these kinds of events as outliers due to their low-spatial density. Furthermore, the last column describes the total amount of energy that is estimated by all possible loads within a day regarding the residual demand.

According to the results, EWH is the only appliance that can be represented by specific loads. Such instances that have been made bold in Table 4 bring about an opportunity to evaluate the identification accuracy of this appliance, as the second major device in Quebec residences, by NILM metrics. Table 5 presents the results of such an examination based on the F1-score and TECA measures [50]. It should be noted that the 7th day is added to this evaluation because of the minor effect of the stove, as explained. In this Table, F1-scores of total outcomes are also presented to estimate the overall constructed sequence, i.e., ON-OFF operation states. The last step is subject to all issues, gathered across the whole process. Therefore, the total results can be expressed as fair for half of the days in Table 5. Nonetheless, the process should be enhanced with regard to all the opportunities and challenges, discussed through the analysis of the case studies. Particularly, the challenges related to blind load disaggregation (the unsupervised step) must be addressed. Load construction from scratch is a difficult task even in high sampling intervals of less than 1 min. Indeed, the thorough investigation, conducted by this study, not only reveals the technical obstacles to developing NILM systems in Quebec but also stimulates the implementation of the proposed combinatory approach of ILM and NILM. The Quebec case is restricted by in-use appliances’ characteristics, data availability, and smart meter properties. Under the suggested load monitoring scheme, suitable data with higher sampling intervals using smart plugs technologies and additional information through HEMS can be supplied to accomplish practical applications.

## 7. Conclusions

This paper provides a thorough investigation into NILM difficulties in Quebec houses. It exploits real-world low-sampling rate data from a set of houses, located in Quebec, to demonstrate the challenging scenario of the case study. It discusses the most important elements of an NILM for actual applications and deepens its exploration by a comparative statistical analysis based on popular public databases. In addition, this work proposes a realistic approach for the case of Quebec by combining the intrusive and non-intrusive aspects of load monitoring. In order to stimulate its suggestion, the study carries out a low-sampling load disaggregation practice to manifest the complications of targeting an NILM-based oriented system for houses in this region. The exercise utilizes a supervised method to extract the power consumption of EBHs in the first step based on deep learning models and, subsequently, employs an unsupervised scheme to identify the power demand of other devices in the remaining load based on clustering techniques. This exhaustive manuscript explores different techniques for every facet of the problem to provide a fruitful discussion with elaborated remarks beneficial to practical implementations. Indeed, the continuation of this study can lead to enhanced load monitoring mechanisms pertinent to the Quebec case.

## Figures and Tables

**Figure 1 sensors-23-07288-f001:**
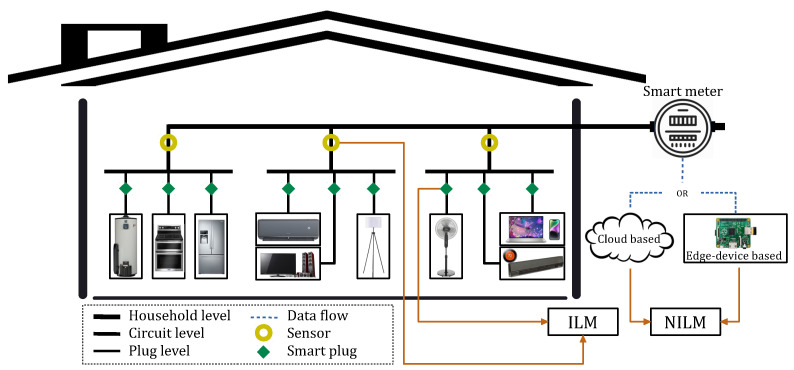
A simple representation of intrusive and non-intrusive approaches to household load monitoring and their technical means [8].

**Figure 2 sensors-23-07288-f002:**
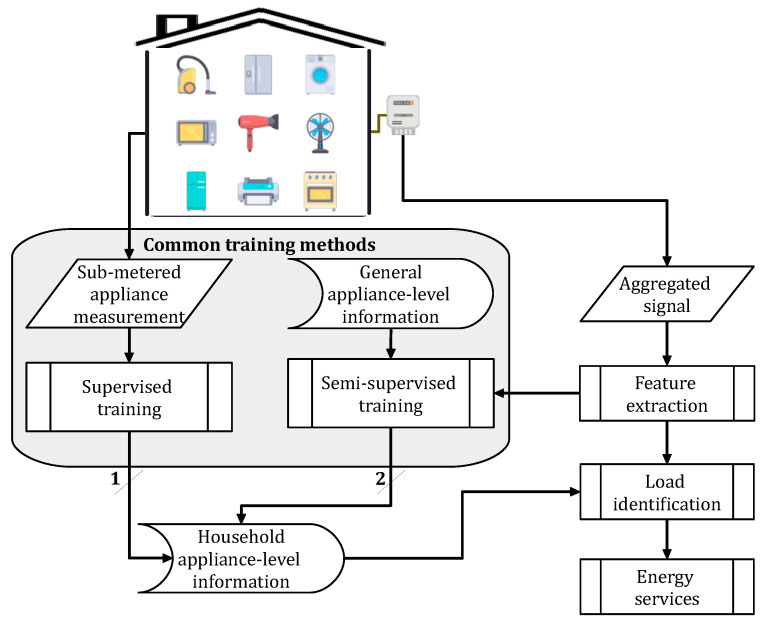
NILM procedure along with its common choice of learning methods practiced by the fundamental research [6].

**Figure 3 sensors-23-07288-f003:**
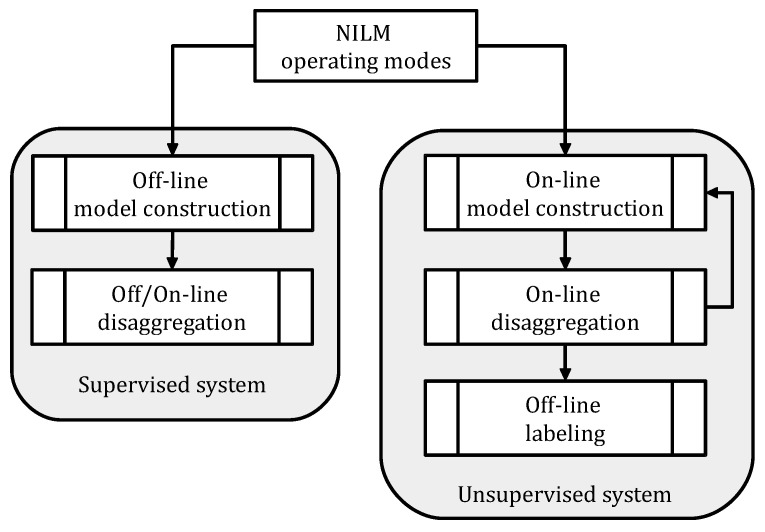
Operation modes of a common NILM system regarding its learning phase [6].

**Figure 4 sensors-23-07288-f004:**
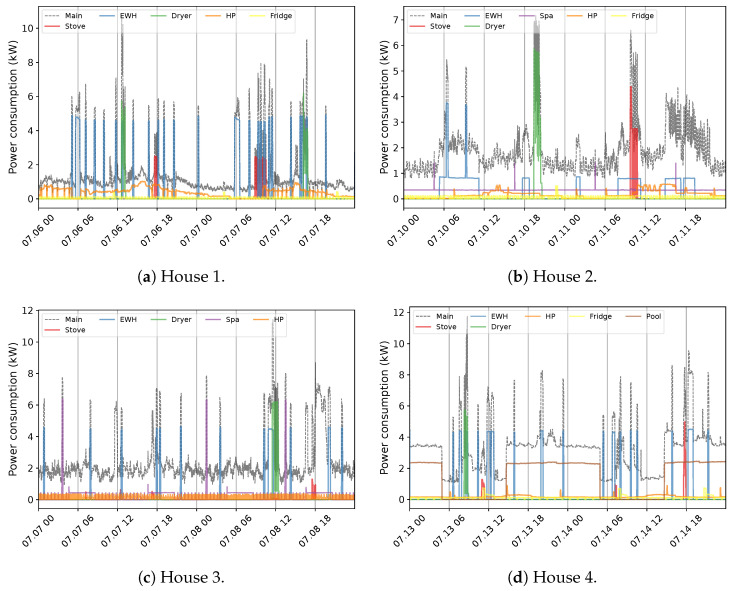
An exemplification of the household power consumption profile in Quebec residences within two days in warm seasons at 1 min sampling intervals.

**Figure 5 sensors-23-07288-f005:**
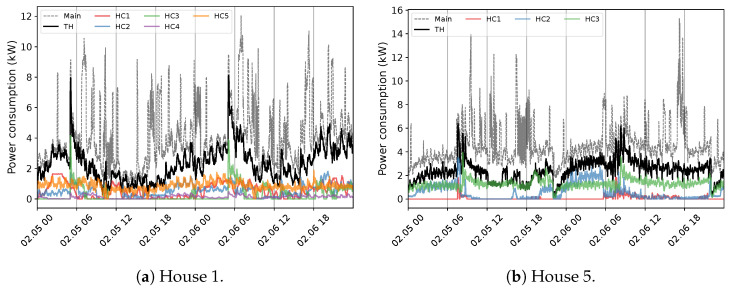
An example of the household heating consumption profile in Quebec residences within two days in cold seasons at 1 min sampling intervals.

**Figure 6 sensors-23-07288-f006:**
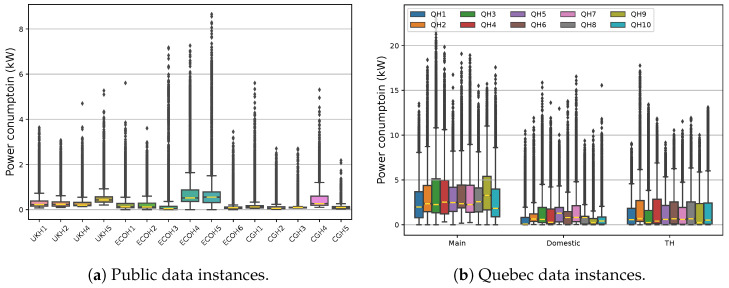
The distribution of public and Quebec data in targeted houses along with the range of the domestic and TH loads from the former.

**Figure 7 sensors-23-07288-f007:**
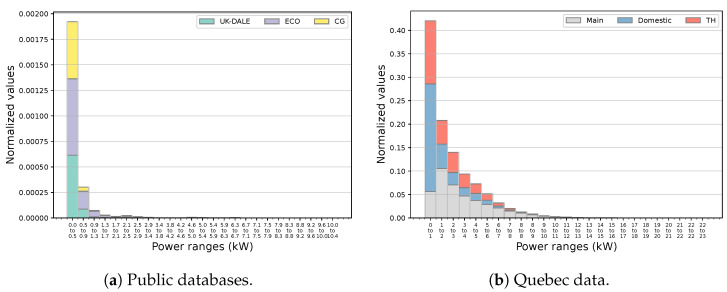
The frequency histogram of public and Quebec data along with relevant domestic and TH shares of the latter.

**Figure 8 sensors-23-07288-f008:**
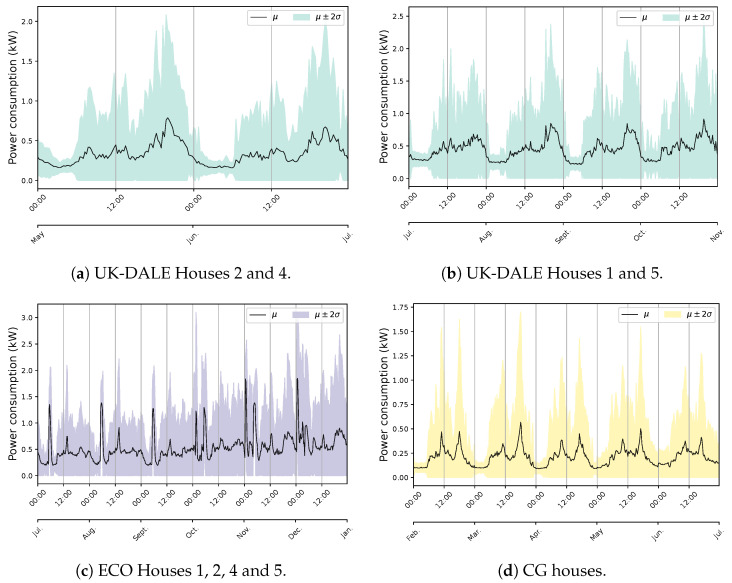
The diurnal behavior of energy consumption in public databases according to data availability and time interval similarity.

**Figure 9 sensors-23-07288-f009:**
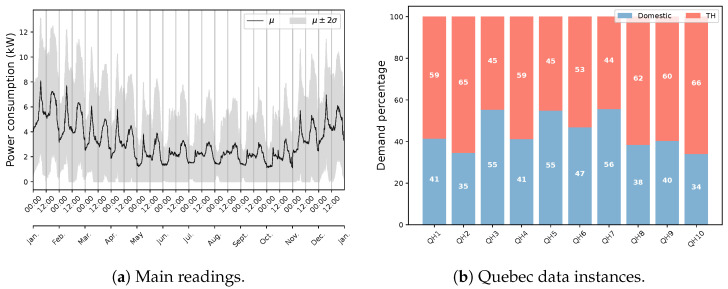
The diurnal behavior of energy consumption in eight Quebec houses according to data available from the main reading.

**Figure 10 sensors-23-07288-f010:**
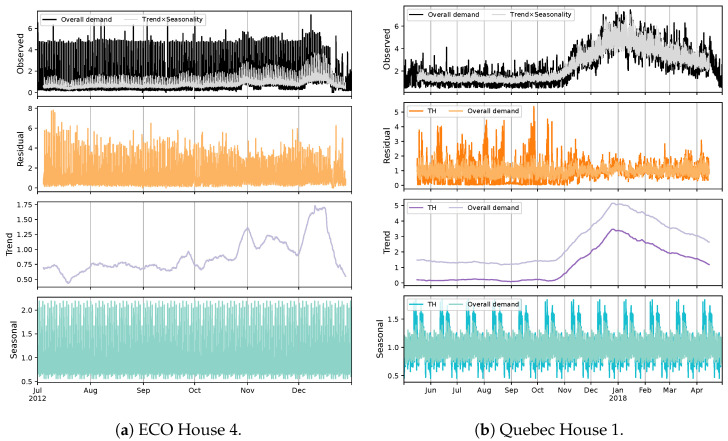
The seasonal decomposition of public and Quebec data for two fine examples based on the multiplicative model.

**Figure 11 sensors-23-07288-f011:**
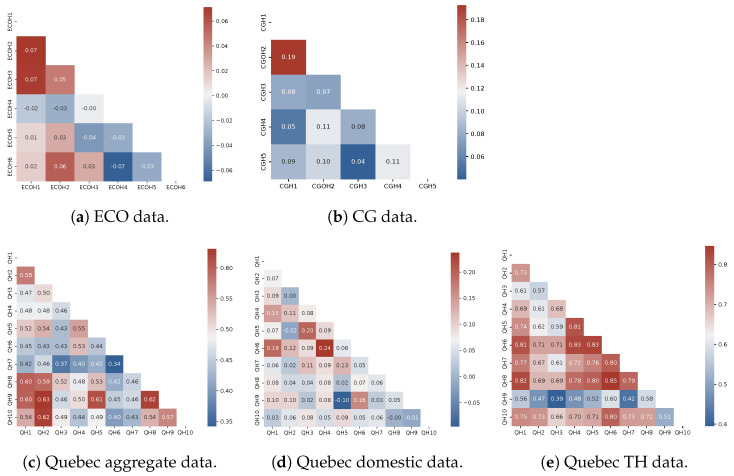
Correlation between available instances in public and Quebec databases along with similar information for domestic and TH loads of the latter.

**Figure 12 sensors-23-07288-f012:**
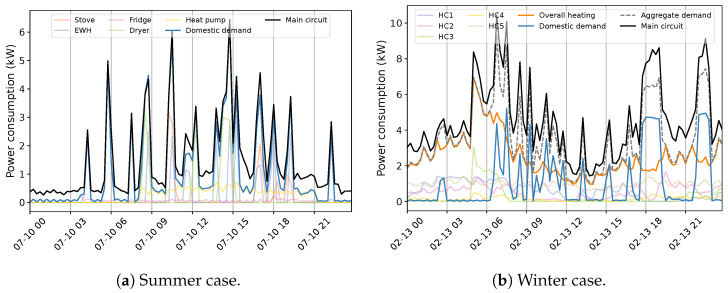
Exemplification of seasonal behavior of power consumption profiles in a Quebec house at a 15 min sampling rate.

**Figure 13 sensors-23-07288-f013:**
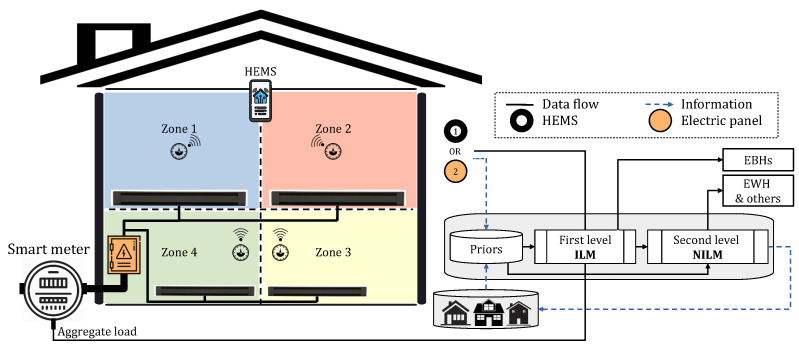
An example of the proposed approach in a house with four thermal zones.

**Figure 15 sensors-23-07288-f015:**
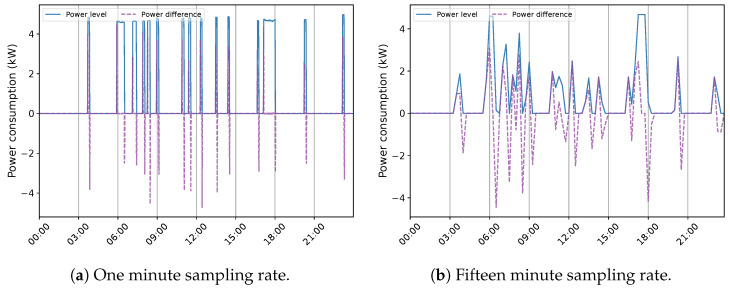
The EWH power profile from Quebec House 1 in (**a**) 1 and (**b**) 15 min sampling time.

**Figure 16 sensors-23-07288-f016:**
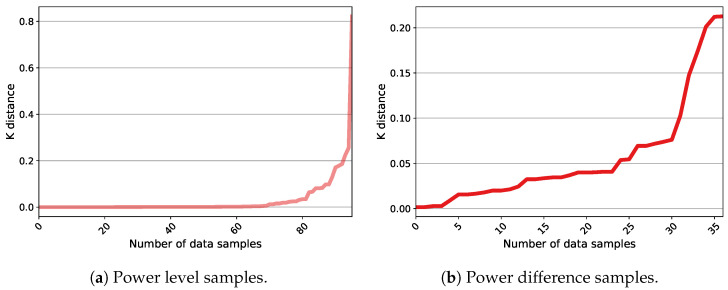
The k-nearest neighbor analysis for one-day data from House 1 with MinPts equal to 4.

**Figure 17 sensors-23-07288-f017:**
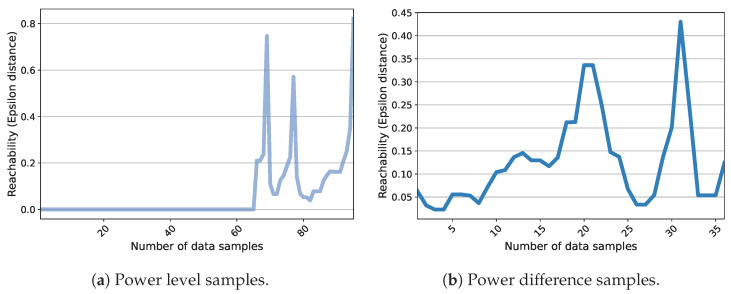
The reachability plot for one-day data from House 1 based on the OPTICS algorithm.

**Table 1 sensors-23-07288-t001:** Ten most cited public datasets for NILM studies in residential houses in descending order [1].

Dataset	Number of Houses	Measuring Duration per House	Sampling Frequency		Site
			**Appliance**	**Aggregate**	
REDD	6	3–19 days	3 s	1 s & 15 kHz	USA
UMass Smart	3	3 months	1 s	1 s	USA
UK-DALE	5	3–17 months	6 s	1–6 s & 16 kHz	UK
BLUED	1	8 days	event label	12 kHz	USA
AMPDs	1	1 year	1 min	1 min	CDN
ECO	6	8 months	1 s	1 s	CH
Tracebase	15	N/A	1–10 s	N/A	DE
HES	251	1–12 months	2–10 min	2–10 min	UK
iAWE	1	73 days	1–6 s	1 s	IND
GreenD	9	1 year	1 s	1 s	AT/IT

**Table 2 sensors-23-07288-t002:** The LSTM results of estimating the overall load of EBHs in Quebec houses.

Data	MAE (kW)	MSE	RMSE (kW)	sMAPE (%)
House 1	0.39	0.41	0.64	46
House 2	1.03	2.68	1.65	88
House 3	0.72	1.71	1.31	56
House 4	0.97	2.26	1.50	76
House 5	0.48	0.69	0.83	46
House 6	0.79	1.37	1.17	58
House 7	0.67	0.99	0.99	69
House 8	0.66	0.69	0.83	54
House 9	0.56	0.66	0.81	45
House 10	0.72	1.62	1.27	72

**Table 3 sensors-23-07288-t003:** The LSTM results of estimating the overall load of EBHs in Quebec House 1.

Day	TECA (%)	MAE (kW)	MSE
1st	86	0.48	0.57
2nd	86	0.41	0.52
3rd	86	0.47	0.37
4th	84	0.53	0.58
5th	84	0.47	0.73
6th	85	0.37	0.36
7th	85	0.38	0.33

**Table 4 sensors-23-07288-t004:** The proposed clustering method results of identifying other loads in residual signal.

Day	Detected Load (kW) and Related Device					
	**1st**	**2st**	**3rd**	**4th**	**5th**	**Energy (%)**
1st	0.92	**3.8** EWH	2.3 EWH Stove	1.9 EWH Stove	**4.6** EWH	30
2nd	0.75	**2.9** EWH	**2.5** EWH	**1.8** EWH	-	37
3rd	0.89	-	-	-	-	-
4th	0.93	2.9 EWH Stove Dryer	4.4 EWH Stove	3.7 EWH Stove	-	38
5th	0.82	3.8 EWH Stove	3.2 EWH Stove Dryer	-	-	23
6th	0.81	**3.1** EWH	-	-	-	34
7th	0.87	**2.9** EWH Stove	**4.9** EWH	-	-	35

**Table 5 sensors-23-07288-t005:** The results of the load identification practice applied to only EWH potential loads.

Day	Load (kW)	F1-Score (%)	TECA (%)
1st	3.8 4.6	25 25	54 55
	Total	44	-
2nd	2.9 2.5 1.8	30 21 39	54 55 59
	Total	62	-
6th	3.1	69	66
7th	2.9 4.9	47 39	57 59
	Total	70	-

## Data Availability

The data for this research is unavailable due to privacy restrictions.

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
