# Peer review of "Towards Feasible Solutions for Load Monitoring in Quebec Residencesâ€"

_sensors, 2023, doi:10.3390/s23167288_

Round 1
Reviewer 1 Report
(1) This paper is not well written, the presentation is obscure and unreadable, for examples, the following words in the abstract such as "the underlying concerns", "the technical specifications", "common matters", "the essence", "public datasets", "specific concerns", "uncommon situations", "pertinent databases", these words are very generalized, the authors should provide the detailed meanings or specified contents to clarify the main idea of this manuscript.
(2) The authors provide Section 3 to study the statistical characteristics of the Quebe data. Is the proposed method only applied on Quebe data? In this context, the reader may think that if one wants to apply the proposed method on other conditions or locations, one needs to do the same process similar to the Section 3 for the Quebe data. If this is true, the proposed method is too difficult to be applied to other conditions and has only limited applications on Quebe data.
(3) The authors are encouraged to reorganize and rewrite this paper for improving the readability. Some concepts and symbols should be defined appropriately if possible, because the reviewer could not find any mathematical symbol in this manuscript, this is strange for a technical paper.
Author Response
The authors would like to thank the reviewer for the valuable and constructive comments. The paper has been revised based on the reviewer’s comments which are discussed below in order.
Reviewer#1
The authors would like to mention that with respect to the general concern of the reviewer with the content clarification (first comment), additional corrections, have been highlighted in violet for further notice.
Reviewer#1, This paper is not well written, the presentation is obscure and unreadable, for examples, the following words in the abstract such as "the underlying concerns", "the technical specifications", "common matters", "the essence", "public datasets", "specific concerns", "uncommon situations", "pertinent databases", these words are very generalized, the authors should provide the detailed meanings or specified contents to clarify the main idea of this manuscript.
Author response:
The paper has been completely revised with regard to the reviewer’s comment. A thorough linguistic review has been performed to clarify the content and remove ambiguity. Particularly, the abstract has been completely modified and the terms emphasized in the remark were clarified or replaced. A specific answer to this comment can be found in Subsection 1.1, pages 2 and 3 (with an additional structure update); Section 4, page 13; and Section 5 page 15.
________________________________________
Reviewer#1, The authors provide Section 3 to study the statistical characteristics of the Quebe data. Is the proposed method only applied on Quebe data? In this context, the reader may think that if one wants to apply the proposed method on other conditions or locations, one needs to do the same process similar to the Section 3 for the Quebe data. If this is true, the proposed method is too difficult to be applied to other conditions and has only limited applications on Quebe data.
Author response:
In fact, the statistical analysis and the load disaggregation practice are separate studies and intend different objectives. The former has been developed to demonstrate the statistical differences between Quebec data and three different public databases. It takes advantage of a set of statistics applied to all four datasets, i.e., Quebec, UK-DALE, ECO, and CeleverGuard (CG). Through this analysis, significant remarks on the potential challenges of Non-Intrusive Load Monitoring (NILM) NILM in Quebec houses are drawn. The latter has been designed to demonstrate the complication of NILM in Quebec houses by popular machine learning methods and, in turn, promote the proposed approach of combining the intrusive aspect of load monitoring (ILM) with the non-intrusive one (NILM). With regard to the reviewer’s comment, the first matter has been clarified in Subsection 3.2, page 8, first paragraph, and page 12, last paragraph; and the second topic has been explained in Section 5, page 15, first paragraph.
________________________________________
Reviewer#1, The authors are encouraged to reorganize and rewrite this paper for improving the readability. Some concepts and symbols should be defined appropriately if possible, because the reviewer could not find any mathematical symbol in this manuscript, this is strange for a technical paper.
Author response:
The paper has been thoroughly reviewed to improve its readability and provide a better understanding of the reviewer’s remark. Particularly, the Motivation in Subsection 1.1 (see pages 2 and 3) has been modified by removing unnecessary explanations, restructuring the paragraphs, and rewriting ambiguous descriptions to clarify the idea behind the whole study. In addition, abbreviations and symbols have been reviewed throughout the paper to remove any existing confusion and relevant references have been provided for the NILM accuracy metrics (see Section 6 pages 18 and 19). Besides, this research utilizes two mathematical concepts. Under a supervised procedure, a Deep Learning (DL) Long Short-Term Memory (LSTM) network and under an unsupervised process, a clustering technique based on DBSCAN (Density-Based Spatial Clustering of Applications with Noise) and OPTICS (Ordering Points To Identify the Clustering Structure) have been designed. On one side, both methods utilize machine learning concepts with well-known mathematical foundations broadly discussed in related fields. Although LSTM is based on the derivation of backpropagation through time, which is quite straightforward, it has an exhaustive mathematical representation. Therefore, it is considered as black box when no contribution to its math is intended. This is the same for other DL models. Subsequently, in NILM research taking advantage of these methods, a configured structure offering the best outcome is contributed and reported in detail. Likewise, DBSCAN and OPTICS utilize basic distance metrics by default Euclidean Distance. As a result, in relevant research, only the mathematical definitions of their input parameters have been given. This has been verified by the author for all the related citations in this study (references 44, 45, and 46 mentioned below). In fact, the proposed clustering approach is reflected by the combination of OPTICS and DBSCAN rather than their underlying math. On the other side, for both concepts, well-known Python libraries have been used, i.e. Keras and Sickit-learn, respectively. Indeed, employing standard procedures based on open-source libraries facilitate technical studies exploring actual implementations. It should be added although these libraries, especially Sickit-learn, provide effective mathematical expressions, it is not the case for the above methods probably due to the aforementioned reasons. Accordingly, this paper has attempted to clarify the mathematical aspects through a concise description while avoiding a verbose repetitive conversation, which is also similar to the tendency of the relevant literature. Notwithstanding, with respect to the reviewer’s comment, the selected LSTM structure has been detailed further (Section 5, page 15, last paragraph), and the values of the DBSCAN and OPTIC parameters have been expressed (Section 5, page 16, last paragraph) for replication ease.
- Deb, C.; Frei, M.; Hofer, J.; Schlueter, A. Automated load disaggregation for residences with electrical resistance heating. Energy and Buildings 2019, 182, 61–74. https://doi.org/10.1016/j.enbuild.2018.10.011.
- Sander, J.; Ester, M.; Kriegel, H.P.; Xu, X. Density-based clustering in spatial databases: The algorithm GDBSCAN and its applications. Data Mining and Knowledge Discovery 1998, 2, 169–194. https://doi.org/10.1023/A:1009745219419.
- Rahmah, N.; Sitanggang, I.S. Determination of Optimal Epsilon (Eps) Value on DBSCAN Algorithm to Clustering Data onPeatland Hotspots in Sumatra. IOP Conference Series: Earth and Environmental Science 2016, 31, 012012. https://doi.org/10.1088/ 1755-1315/31/1/012012.
Reviewer 2 Report
the paper presented good work but there are some concerns:
1- The paper is well written.
2- The authors mentioned deep learning and machine learning for the proposed model, but it is not clear which algorithm is implemented and how this help additionally the figures presented are figure from experimental results. Can you justify this please.
3- Also, in the conclusion you didn't mention the implementation of deep learning as you did in the abstract.
Author Response
The authors would like to thank the reviewer for the valuable and constructive comments. The paper has been revised based on the reviewer’s comments which are discussed below in order.
Reviewer#2
The authors would like to mention that with respect to the general concern of the reviewer with the content clarification (second comment), additional corrections, have been highlighted in violet for further notice.
Reviewer#2, The paper is well written.
Author response:
The authors would like to extend their thanks to the reviewer for the encouraging comment.
________________________________________
Reviewer#2, The authors mentioned deep learning and machine learning for the proposed model, but it is not clear which algorithm is implemented and how this help additionally the figures presented are figure from experimental results. Can you justify this please.
Author response:
The paper has been thoroughly revised with regard to reviewer’s comment to clarify the idea, detail the mathematical aspects, and remove any ambiguous explanation. In fact, the proposed algorithm takes advantage of a supervised method based on Deep Learning (DL) Long Short-Term Memory (LSTM) networks, and an unsupervised technique based on a combination of DBSCAN (Density-Based Spatial Clustering of Applications with Noise) and OPTICS (Ordering Points To Identify the Clustering Structure) clustering schemes. The promise of LSTM in capturing long-term dependencies in time-series is used to extract the overall demand of Electric Baseboard Heaters (EBHs) from the household aggregate power consumption. Subsequently, the application of DBSCAN to spatial clustering elaborated by OPTICS is utilized to identify other potential appliances, specifically Electric Water Heaters (EWHs) in the remaining signal. This strategy allows to tackle the Non-Intrusive Load Monitoring (NILM) concept in Quebec residences and reveal its complications regarding actual implementations. Consequently, it promotes the proposed solution that relies upon integrating the intrusive approach of load monitoring (ILM) with the non-intrusive one to alleviate its difficulties. The answer to this remark can be found in page 15, Figure 14 (the proposed method) and last paragraph (the LSTM architecture), and page 16, last paragraph (the clustering mechanism). Throughout the paper, all Figures and Tables are the outcomes of different algorithmic procedures applied to actual data. The statistical analysis has been carried out by use of real-world data from Quebec residences as well as UK-DALE, ECO, and CeleverGuard (CG) databases. The proposed NILM has been experimented with Quebec data (the former). This concern has been answered in Section 3, page 7, Subsection 3.2, page 8, first paragraph, and Section 6, page 18, first paragraph.
________________________________________
Reviewer#2, Also, in the conclusion you didn't mention the implementation of deep learning as you did in the abstract.
Author response:
The paper has been reviewed to improve the Conclusion Section through a sufficient description regarding the reviewer’s remark. The answer to this comment can be seen in Section 7, page 21.
Reviewer 3 Report
The paper presents applied research on the NILM with a focus on Quebec residences. The authors emphasize that implementing NILM systems in real-world residential settings requires overcoming technical challenges and addressing concerns beyond what has been explored in basic research using public datasets.
The authors perform a strong statistical analysis, utilizing real-world measurement data from Quebec and European regions. This analysis highlights the specificity and potential challenges of the case study in alignment with NILM requirements.
For the experimental setting, the authors use a combinatorial approach for load identification. This approach leverages sub-meter smart technologies to integrate both intrusive and non-intrusive aspects of load monitoring. The intrusive aspect extracts overall demand from the aggregate load using standard deep learning models, while the non-intrusive aspect disaggregates the residual load through unsupervised clustering techniques.
Overall, the paper is well-written and contributes to the understanding of NILM in the residential sector, particularly in the specific context of Quebec houses. However, I have some concerns listed below.
1) I recommend some revisions to the abstract. While it effectively summarizes the content, it appears to be too long. I think that condensing the abstract to a more concise form could enhance its readability and effectiveness.
2) The authors provide a literature review of existing approaches for low-rate NILM, which includes variants of Hidden Markov Models and Deep Neural Networks, but only briefly mention combinatorial and optimization approaches. A substantial part of the missing literature comprises linear programming and mixed integer non-linear programming approaches (either event-based or state-based) for low-rate NILM. See these recent papers:
[1] https://doi.org/10.1109/TCSII.2016.2603479
[2] https://doi.org/10.1109/TSG.2022.3152147
3) The paper provides a detailed discussion of the parameter settings used in their experiments. However, it would be helpful if the authors could report the percentage of noise represented by unknown loads for each dataset.
Minor editing of English language required
Author Response
The authors would like to thank the reviewer for the valuable and constructive comments. The paper has been revised based on the reviewer’s comments which are discussed below in order.
Reviewer#3
The authors would like to mention that with respect to the general concern of the reviewer with the content clarification (first comment), additional corrections, have been highlighted in violet for further notice.
Reviewer#3, I recommend some revisions to the abstract. While it effectively summarizes the content, it appears to be too long. I think that condensing the abstract to a more concise form could enhance its readability and effectiveness.
Author response:
The paper has been revised with regard to the reviewer’s comment and the abstract has been completely updated to avoid a verbose explanation. The answer to this remark can be found on page 1.
________________________________________
Reviewer#3, The authors provide a literature review of existing approaches for low-rate NILM, which includes variants of Hidden Markov Models and Deep Neural Networks, but only briefly mention combinatorial and optimization approaches. A substantial part of the missing literature comprises linear programming and mixed integer non-linear programming approaches (either event-based or state-based) for low-rate NILM. See these recent papers:
[1] https://doi.org/10.1109/TCSII.2016.2603479
[2] https://doi.org/10.1109/TSG.2022.3152147.
Author response:
The paper has been modified regarding the sensible concern of the reviewer. Particularly, Section 2.1 on page 4 has been updated to cover optimization-based methods and cite relevant articles. It should be mentioned that since this manuscript provides an exhaustive investigation, the NILM concept has been briefly reviewed to point out its essential elements, which are supported by related references for further studies.
________________________________________
Reviewer#3, The paper provides a detailed discussion of the parameter settings used in their experiments. However, it would be helpful if the authors could report the percentage of noise represented by unknown loads for each dataset.
Author response:
The paper has been modified with respect to the reviewer’s comment. In fact, an important objective of this study is to reveal the obstacles to actual load monitoring implementations, particularly NILM, in Quebec residences. In this regard, our analyses show that the unsupervised process of the method should be improved in future work since as stated, it results in fair outcomes only for Electric Water Heater (EWH). For this reason, the study has not been numerically detailed about the issues brought about by unknown demand. In fact, the literature applied to public databases has the same tendency about this type of load since it is difficult to treat (no label for evaluation purposes). Notwithstanding, the share of this type of loads is very considerable in Quebec houses regarding the power consumption ranges. Respectively, 26%, 26%, 22%, 21%, 16%, 21%, 49%, and 38% of the load in Houses 1, 2, 3, 4, 5, 6, 7, and 9 belong to unknown usage, considered as a disturbance. It should be noted there is no ‘MAIN’ circuit record for Houses 8 and 10 as pointed out in the paper. This comment has been briefly clarified in Section 6, page 19, first paragraph.
Round 2
Reviewer 1 Report
All my comments have been carefully and effectively addressed.
Reviewer 2 Report
The authors have addressed the required comments.
Reviewer 3 Report
The authors have addressed all my comments.
-